The Aurora kinase inhibitor AT9283 inhibits Burkitt lymphoma growth by regulating Warburg effect

Jiang Kaiming 1
Bai Lihong 1
Wang Canfei 1
Xiao Xiang 1 2
Cheng Zhao 1 2
Peng Hongling 1 2
Liu Sufang sufang-liu@csu.edu.cn 1 2
1 Department of Hematology, The Second Xiangya Hospital, Central South University , Changsha , Hunan , China
2 Institute of Molecular Hematology, Central South University , Changsha , Hunan , China
Vassetzky Yegor
Electronic publication date: 2023 Dec 11
Publication date: 2023
Volume: 11
Electronic Location ID: e16581
Received 2023 Aug 23; Accepted 2023 Nov 13
Copyright: ©2023 Jiang et al.
Copyright year: 2023
Copyright holder: Jiang et al.
License: This is an open access article distributed under the terms of the Creative Commons Attribution License, which permits unrestricted use, distribution, reproduction and adaptation in any medium and for any purpose provided that it is properly attributed. For attribution, the original author(s), title, publication source (PeerJ) and either DOI or URL of the article must be cited.
License URL: https://creativecommons.org/licenses/by/4.0/

Keywords: Aurora kinase, AT9283, Burkitt lymphoma, Warburg effect, c-MYC, HIF

Funding: Natural Science Foundation of Hunan Province 2021JJ30919 Changsha Municipal Natural Science Foundation kq2014252 This work was supported by the Natural Science Foundation of Hunan Province (No. 2021JJ30919) and the Changsha Municipal Natural Science Foundation (No. kq2014252). The funders had no role in study design, data collection and analysis, decision to publish, or preparation of the manuscript.

==============================
Objective

To investigate the effect of the kinase inhibitor AT9283 on Burkitt lymphoma (BL) cells and elucidate the underlying mechanisms.

Methods

The effect of AT9283 on the proliferation of BL cell lines was tested using the MTT assay. Apoptosis and cell cycle were measured by flow cytometry. The proteins associated with the cell cycle, apoptosis, and the Warburg effect were detected using Western blotting. Alterations in glycolytic metabolism in terms of glucose intake and lactate concentrations were determined by glucose and lactate assays.

Results

The current study utilized the GEPIA, the Human Protein Atlas (HAP) database and immunohistochemistry to conduct analyses, which revealed a high expression of Aurora kinases and Warburg effect-related proteins in malignant B-cell lymphoma tissues. AT9283 significantly inhibited the cell proliferation of BL cells and induced G2/M arrest. Additionally, AT9283 induced apoptosis in BL cells and reversed the Warburg effect by increasing glucose uptake and reducing lactate production. Moreover, the protein expression of hexokinase 2, pyruvate kinase M2, and lactate dehydrogenase A was significantly suppressed by AT9283, possibly through the inhibition of c-Myc and HIF-1α protein expression.

Conclusion

The reversal of the Warburg effect in BL cells and the subsequent inhibition of cell proliferation and induction of apoptosis were observed by targeting Aurora A and Aurora B with AT9283. This finding may present new therapeutic options and targets for BL.

Introduction

Burkitt lymphoma (BL) is a B-cell non-Hodgkin’s lymphoma (B-NHL) that is highly invasive and characterized by rapid proliferation (Mrdenovic et al., 2019; Roschewski, Staudt & Wilson, 2022). This malignancy primarily affects children and young adults, with a predilection for the maxillofacial, central nervous system, abdominal organs, and other extranodal sites. In developing countries, BL represents the majority of B-NHL cases in children. In developing countries, BL accounts for approximately 80% of B-NHL cases in children (Egan, Goldman & Alexander, 2019). Adults and immunocompromised individuals experience less favorable outcomes and more pronounced adverse effects from treatment compared to pediatric patients (Dunleavy, Little & Wilson, 2016; Perkins & Friedberg, 2008). While the initial treatment for Burkitt lymphoma is notably efficacious, the vast majority of patients who experience recurrence or resistance to treatment ultimately succumb to the disease. Retrospective population-based studies have suggested that the incidence of treatment failure among adult patients with Burkitt lymphoma may be as high as 35% (Evens et al., 2021). Consequently, there is an urgent need for the development of novel treatment modalities and intervention strategies to enhance the therapeutic outcomes for this malignancy.

The degree of tumor malignancy depends on the growth efficiency of tumor cells, and an adequate energy supply is the prerequisite and necessary condition for the malignant proliferation of tumor cells. The Warburg effect refers to the phenomenon in which although the oxygen level is sufficient to support mitochondrial oxidative phosphorylation, tumor cells still metabolize glucose by glycolysis (Cantor & Sabatini, 2012). This low-efficiency mode of metabolism in cancer cells results in the consumption of a large amount of glucose compared with that in normal cells. The lactic acid secreted can not only inhibit the formation of an immunosuppressive microenvironment but also allow rapid synthesis of the intermediate products required for proliferation. Therefore, targeting metabolic links to tumor glycolysis is a key strategy in current antitumor treatment.

The family of Aurora kinases is a group of serine/threonine kinases widely expressed in a variety of cells and has three members, Aurora A, Aurora B, and Aurora C, which play important roles in mitosis, chromosome segregation, and spindle formation in eukaryotic cells (Keen & Taylor, 2004). In hematological malignancies, overexpression of Aurora A and Aurora B is found in patients with cytogenetic abnormalities indicating an unfavorable prognosis. A previous study discovered an important role for Aurora-A in glycolytic modulation, as well as a mechanism through which LDHB contributes directly to the Warburg effect (Cheng et al., 2019). Further, inhibition of Aurora A and Aurora B is an attractive anticancer approach for a variety of cancers, including solid tumors and hematological malignancies (Gavriilidis, Giakoustidis & Giakoustidis, 2015; Ikezoe et al., 2007; Rodrigues Alves et al., 2016).

AT9283 is a multikinase inhibitor discovered using a fragment-based approach and is highly active against Aurora kinases, Abl, and JAK (Howard et al., 2009). A number of studies have shown that AT9283 has therapeutic potential for leukemia, myeloproliferative disorders, aggressive B-cell lymphomas, and colon cancer (Qi et al., 2012; Takeda et al., 2020; Torrente et al., 2020). So far as we know, there exists no report regarding the impact of AT9283 on the Warburg effect in BL. In this study, we aimed to explore the effects of AT9283 on the survival of Burkitt lymphoma cells, including primary Burkitt lymphoma cells, and to elucidate the underlying molecular mechanisms. Our research has revealed that AT9283 demonstrates noteworthy anti-BL properties through the inhibition of cellular proliferation, the induction of apoptosis, and the arrest of the cell cycle. The mechanism is linked to the reversal of the Warburg effect by means of a reduction in the expression of c-MYC and HIF1α proteins.

Materials & Methods

Reagents and antibodies

AT9283, which was acquired from Selleckchem, was dissolved in DMSO to create a stock solution of 10 mmol/L. MTT was dissolved in PBS and stored at a temperature of −20 °C. Antibodies against cleaved PARP (#9541), Caspase 3 (#9665), Cyclin B1 (#12231), c-Myc (#9402), Aurora A (#12100p), Aurora B (#3094), and HK2 (#2867) were procured from Cell Signaling Technology. Antibodies against β-actin were obtained from Sigma, while antibodies against PKM2 (A0268) were obtained from ABclonal. The antibody against HIF-1α (20960-1-AP) was procured from Proteintech.

Patient sample, cell lines and reagents

The Burkitt lymphoma cell lines P3HR1 and Akata were generously provided by Prof. Y. Cao of Central South University in Changsha, China. Bone marrow mononuclear cells (BMMNCs) were isolated from a 61-year-old male patient with Burkitt leukemia variant, within our department (Leoncini et al., 0000; Zhang et al., 2021) in our department. Both cell lines and primary BMMNCs were cultured using standard cell culture techniques. The study was granted approval by our Ethical Committee (the Ethical Committee of Xiangya Hospital, Central South University), and the subject provided written informed consent (Approval No. 2021772).

Detection cell proliferation

As described previously (Zhang et al., 2021), the AT9283’s effects on cell viability was detected using the MTT assay. Briefly, cells were seeded in triplicate wells of 96-well microplates at a density of 5 × 104 cells per well and incubated with varying concentrations of AT9283 for 24 or 48 h. The percentage of viable cells was determined by comparing the AT9283-treated group to an untreated control group. The dose–response curve was calculated using GraphPad sofware with the nonlinear regression curve fit.

Analyzing cell apoptosis and cycle with flow cytometry

Akata and P3HR1 cells were subjected to varying concentrations of AT9283 for a duration of 48 h. Following treatment, the cells were harvested and subjected to centrifugation at 1,000 rpm for a duration of five minutes. A fixative solution containing 70% ethanol was utilized to fix the cells overnight at a temperature of 4 °C for the purpose of cell cycle analysis. Subsequently, flow cytometry was performed on the cells after staining with Propidium Iodide (PI) (KeyGEN BioTECH, Nanjing, China). Annexin V and PI (Multisciences, China) staining was performed on the cells to analyze apoptosis, and the cells were counted using a FACSCalibur instrument (BD FACSCalibur; Becton-Dickinson, Brea, CA, USA).

Western blot analysis

As described previously (Xiao et al., 2019), total protein was prepared and immunoblot analysis was performed. Briefly, following treatment with AT9283 for 24–48 h, cells were lysed on ice using lysis buffer and proteins were subsequently collected via centrifugation. After quantifying the protein concentrations, SDS-PAGE was utilized to separate the proteins, which were then transferred to PVDF (polyvinylidene fluoride) membranes. The membranes were blocked with 5% defatted milk for 1 h at room temperature, followed by overnight incubation with primary antibodies at 4 °C and secondary antibodies conjugated with HRP for 1 h at room temperature. The visualization of immunoreactions was achieved through the utilization of a chemiluminescence detection system. The analysis of resulting images was conducted using the Gel DocTM XRS system (Bio-Rad, Hercules, CA, USA), while the quantification of relative protein levels was performed using Image J Software.

Immunohistochemical (IHC) staining

The IHC staining procedure was conducted according to our previous descriptions (Liu et al., 2013), In summary, paraffin sections were subjected to dewaxing and dehydration using xylene and gradient ethanol, followed by immersion in a boiling sodium citrate buffer (10 mM, pH 6.0) for 10 min to retrieve antigens. Endogenous peroxidase activity was blocked by adding a 3% H2O2 solution to the slides for 15 min. The slides were then washed with PBS and blocked with 10% serum albumin in a humidified chamber at room temperature for 1 h. The primary antibody was incubated with the slides at 4 °C overnight, followed by hybridization with the secondary antibody for 45 min to visualize the target protein using DAB substrate. The sections were subsequently stained with hematoxylin and fixed with neutral gum. The staining results were then observed and analyzed under a microscope in accordance with academic standards.

GEPIA and Human Protein Atlas (HPA) database analysis

The Gene Expression Profiling Interactive Analysis (GEPIA) is a web-based database and mining tool that offers interactive gene expression analysis and profiling for both cancerous and normal genes (http://gepia.cancer-pku.cn/). The database employs a nonlogarithmic scale for calculation purposes, while a log-scale axis is utilized for visualization. The normalization of mRNA levels in this tool is achieved through the use of Transcripts per Million (TPM) and unpaired Student’s t tests (Tang et al., 2019; Tang et al., 2017b). More than 24,000 human proteins in tissues and cells are represented in the the HPA database (https://www.proteinatlas.org). Thus, we searched the HPA website to analyze Aurora A and Auror B expression in the different cancer and non-cancerours type cell lines available on this website.

Flow cytometry dectection of cleaved PARP and active caspase-3

The evaluation of cleaved PARP and active caspase-3 in BL cells treated with AT9283 was conducted through the utilization of fluorescently labeled anti-active caspase-3 (#550914, BD Pharmingen) and anti-cleaved PARP (#558710; BD Pharmingen, Franklin Lakes, NJ, USA) antibodies. The experimental procedures were executed in accordance with the instructions provided by the reagent. in summary, the cells were collected and subjected to a 30-minute incubation in BD Cytofix/Cytoperm™ solution to enhance the permeability of the cell membrane. Subsequently, the cells were washed and incubated with the aforementioned antibodies. Following the washing process, the cells were reconstituted in PBS and subjected to flow cytometric analysis utilizing a FACSCalibur instrument (BD).

Glucose uptake, lactate production and HK2 activity assays

Following treatment with AT9283 (0, 1, 2, 4 μM) for 24 or 48 h, glucose uptake, lactate production, and HK2 activity were assessed in Akata and P3HR1 cells using commercial Kits from Nanjing Jiancheng Bioengineering Institute (No. F006-1-1 and A019-2-1, Nanjing, China) in accordance with the manufacturer’s instructions. Absorbance was measured using a microplate reader.

Quantitative reverse transcription polymerase chain reaction

RT-PCR was performed as described previously (Zhang et al., 2021). The complete procedure and all the primers information are to be found in the Supplementary Methods.

Statistical analysis

The statistical analysis of the results was conducted using GraphPad Prism. The data from three independent experiments were presented as mean ± SEM values, and the significance of any observed statistical differences was determined through the application of one-way ANOVA and T tests. A P value of less than 0.05 was considered indicative of a statistically significant difference.

Results

AT9283 suppresses the BL cell lines and primary cells proliferation

In order to verify the inhibitory effects of AT9283 on the constitutive activation of Auroras in BL cells, the expression levels of Aurora A and B were assessed in cells treated with AT9283. The results obtained from western blotting analysis indicated that AT9283 significantly down-regulated the protein expression of Aurora A and Aurora B in a dose-dependent manner as depicted in Fig. 1A.

Figure 1 AT9283 suppresses the proliferation of BL cells.

(A) The protein expression of Aurora A and B in BL cells treated with AT9283 for 48 h was performed by Western blot analysis. In the comparison with the untreated group, the intensity of bands was quantified and analyzed by t test using actin as an internal control, densitometric quantitation of protein expression levels are shown as fold changes. Data represented the mean ± s.d. of three independent experiments. (B) Akata, (C) P3HR1 and (D) primary BL cells were treated with different concentrations of AT9283 at indicated time, cell viability was assessed by MTT. **p < 0.01, ***p < 0.001.

The Akata and P3HR1 cell lines were subjected to varying concentrations of AT9283 (0, 1, 2, or 4 μmol/L) for 24 or 48 h, followed by analysis of cell viability using the MTT assay. The results demonstrated a dose- and time-dependent reduction in cell viability of both Akata and P3HR1 cells upon treatment with AT9283 (Figs. 1B, 1C and Fig. S1). Furthermore, the antiproliferative effect of AT9283 on primary BL cells was evaluated, and the MTT assay revealed a dose-dependent inhibition of viability in BMMCs isolated from newly diagnosed BL patients (Fig. 1D).

AT9283 induces cell cycle arrest in BL cells

In order to further investigate the antiproliferative mechanism in BL cells of AT9283, we conducted a flow cytometry analysis of its effect on cell cycle progression. Our findings revealed a significant increase in the percentage of G2/M-phase and a decrease in the percentage of S-phase in Akata and P3HR1 cells with increasing concentrations of AT9283. These results suggest that AT9283 effectively inhibits BL cell proliferation by inducing G2/M-phase blockage (Fig. 2A). Additionally, Western blotting analysis revealed a downregulation of cyclin B1 protein expression, which may play a role in the regulatory mechanism of AT9283-induced cell cycle arrest in BL cells (Fig. 2B).

Figure 2 AT9283 induces G2/M cell cycle arrest of BL cells.

(A) Analysis of the cell cycle was determined by flow cytometry assay in Akata and P3HR1 cells treated with AT9283 for 48 h. (B) The protein levels of cyclin B1 in BL cells treated with the indicated concentrations of AT9283 for 48 h were detected by western blot analysis, and measured using densitometry. ***p < 0.001 and ****p < 0.0001.

AT9283 induces apoptosis via a caspase-dependent pathway in BL cells

Due to the close correlation between cell cycle and apoptosis. Subsequently, the induction of apoptosis was assessed in BL cells treated with AT9283 for 48 h using annexin V-FITC/PI staining. The findings demonstrated that AT9283 elicited apoptosis in a dose-dependent manner, as illustrated in Figs. 3A and 3B. The induction of apoptosis by AT9283 was further evidenced by the increases in the levels of apoptotic markers. Flow cytometry and western blotting were carried out to assess the protein levels of caspase-3 and cleaved PARP. The results showed in Akata and P3HR1 cells treated with AT9283, the procaspase-3 level was reduced (Fig. 3E), while the levels of caspase3 and PARP were increased with AT9283 of dose (Figs. 3C, 3D and 3E). These findings suggest that AT9283 activates the mitochondrial caspase cascade.

Figure 3 AT9283 induces the apoptosis of BL cells in a caspase-dependent pathway.

(A) Apoptotic cells were detected by by Annexin-V/PI double labeling flow-cytometry in Akata and P3HR1 cells treated with AT9283 for 48 h. (B) The percentages of the annexin-V positive (apoptosis) cells were calculated and presented as the graph. (C) flow cytometry analysis was used to detect the expression of actived caspase-3 and cleaved PARP. (D) The graphs present the percentages of cleaved PARP and actived caspase-3 positive cells. (E) Effects of AT9283 on pro-Caspase-3, cleaved caspase-3 and cleaved PARP protein expression (western blot). Densitometry was used to measure the protein levels. ***p < 0.001 and ****p < 0.0001.

Aurora A, B and Warburg effect-related proteins is overexpressed in B cell lymphoma

The reprogramming of glucose metabolism leading to an augmented aerobic glycolysis, commonly known as the Warburg effect, is recognized as a distinctive feature of tumor cells (Fukushi et al., 2022), and plays a crucial role in tumor progression. Lactic acidosis has been reported as the most common types of acidosis in many malignant tumors, including hematological malignancies, especially lymphomas (Looyens et al., 2021; Sanivarapu et al., 2022). To ascertain the involvement of glycolysis in the progression of B-cell lymphoma, we conducted a comparative analysis of the mRNA expression levels of Aurora A and Warburg effect-related genes in B-cell lymphoma and normal tissues using the GEPIA online database. Results demonstrate that Aurora A, Aurora B, and the Warburg effect-related genes LDHA (lactate dehydrogenase A) and PKM (pyruvate kinase M) were significantly overexpressed in tissues obtained from patients with diffuse large B-cell lymphoma (DLBCL) as compared to normal controls (Figs. 4A and 4B). We also analyzed the HPA database to evaluate the Aurora A and B mRNA levels in different cell lines, and revealed that the basal expression levels of Aurora A and B in lymphoma cell lines were high (Fig. S2). Furthermore, we conducted immunohistochemical (IHC) staining to compare the protein expression of PKM and LDHA between the lymph node of Burkitt lymphoma (BL) and normal controls. Our results revealed that the expression of PKM and LDHA proteins was higher in BL lymph node tissues than in normal tissues, which is in agreement with the GEPIA analysis findings (Fig. 4C).

Figure 4 Aurora A, B and Warburg effect-related proteins is overexpressed in B cell lymphoma.

(A, B) Using GEPIA (Gene Expression Profiling Interactive Analysis) dataset (http://gepia.cancer-pku.cn/), we compared the mRNA expression of Aurora A, B and Warburg effect-related LDHA, PKM proteins between diffuse large B cell lymphoma (DLBCL) and normal samples. (C) PKM and LDHA expression in BL and normal samples were examined by IHC. *p < 0.05.

AT9283 attenuates the Warburg effect in BL cells

Nguyen et al. have demonstrated that the inhibition of Aurora kinase A can effectively reverse the Warburg effect in glioblastomas and result in unique metabolic vulnerabilities (Nguyen et al., 2021). Then AT9283 was used to test glucose uptake and lactate production in BL cells in order to confirm whether Aurora kinase inhibition reverses the Warburg effect. We measured glucose and lactate (LA) in cell culture medium and found that AT9283 significantly decreased glucose uptake and LA production in the treated groups compared with the untreated groups in both Akata and P3HR1 cells (Figs. 5A, 5B). These results indicate that AT9283 effectively reverses the Warburg effect. Furthermore, our findings demonstrate that the protein levels of the crucial glycolytic enzymes HK2 and PKM2 were dose dependently decreased in the AT9283-treated groups in both Akata and P3HR1 cells (Figs. 5C, 5D).

Figure 5 AT9283 attenuates the Warburg effect in BL cells.

Akata and P3HR1 cells were treated by AT9283. Then glucose uptake (A), lactate production (B) and activity of HK (C) were assessed at 24 h. (D) AT9283 inhibits glycolytic related proteins in BL Akata and P3HR1 cells. BL cells were treated with AT9283 for 48 h. Expression of HK2 and PKM2 was detected by immunoblotting. (E) The protein levels were calculated and presented as a ratio relative to β-actin. *p < 0.05, **p < 0.01, **p < 0.01.

AT9283 inhibits c-Myc and HIF-1a expression involved in the regulation of glycolytic activity in BL cells

The robust glycolytic activity observed in tumor cells is closely associated with the dysregulation of signaling molecules, particularly the aberrantly activated hypoxia-inducible factor 1-alpha (HIF1α) and c-Myc. To ascertain the inhibitory effects of AT9283 on the expression of c-Myc and HIFα proteins in Burkitt lymphoma cells, western blotting was employed to detect protein expression in cells treated with AT9283. The results indicate that AT9283 effectively suppresses c-Myc and HIF-1α protein expression in a dose-dependent manner (Fig. 6A).

Figure 6 AT9283 inhibits c-Myc and HIF-1a protein expressions in BL cells.

Akata and P3HR1 cells were treated with AT9283 for 48 h. (A) protein expressions of c-Myc and HIF-1a was detected by immunoblotting. (B) The protein levels were calculated and presented as a ratio relative to β-actin. * p < 0.05, ***p < 0.001, ****p < 0.0001.

Discussion

Aurora kinases, which are members of the serine/threonine kinase family, are localized at the two poles of the mitotic spindle during mitosis and are are referred to as aurora protein kinases. The family is divided into three members, namely AURKA/B/C, based on their structural and functional homology (Aurora kinase A, B, C) (Keen & Taylor, 2004), and they play crucial roles in regulating the functions of centrosomes and microtubules during cell mitosis (Adams, Carmena & Earnshaw, 2001). Aurora kinases are frequently over-expressed or abnormally activated in various types of solid tumors with poor prognosis, including colorectal, prostate, lung, ovarian, breast, gastric, liver, and oral cancers, as well as in lymphoma and other hematologic malignancies (de Mel et al., 2019; Murga-Zamalloa, Inamdar & Wilcox, 2019; O’Connor et al., 2019; Tang et al., 2017a; Ulisse, 2017). Aurora kinases play a crucial role in ensuring accurate progression through mitosis, while also mediating a diverse range of signal transduction pathways (Bertolin & Tramier, 2020; Bodvarsdottir, Vidarsdottir & Eyfjord, 2007; Donnella et al., 2018; Ishikawa et al., 2011; Katayama et al., 2004), that impact cell survival, cell cycle arrest, apoptosis resistance, angiogenesis, radio/chemoresistance, tumor recurrence and metastasis, cell aging, and other biological behavioral alterations (Dominguez-Brauer et al., 2015; Yan et al., 2016). Consequently, Aurora kinases are considered promising therapeutic targets (de Mel et al., 2019; Goldenson & Crispino, 2015; Murga-Zamalloa, Inamdar & Wilcox, 2019; O’Connor et al., 2019; Tang et al., 2017a; Ulisse, 2017).

AT9283, a pyrazole-benzimidazole derivative, is a multikinase inhibitor that can effectively inhibit the activity of AURKs and several other kinases, including JAKs, Flt3, and Abl (Howard et al., 2009). This inhibitor can affect the growth and survival of multiple types of cancers, such as multiple myeloma, chronic myelocytic leukemia (CML), lymphomas, and colorectal cancer (Curry et al., 2009; Qi et al., 2012; Santo et al., 2011; Takeda et al., 2020; Tanaka et al., 2010). Additionally, AT9283 as a drug for multiple myeloma has already entered the clinical stage (Hay et al., 2016) and leukemia (Vormoor et al., 2017). Research has demonstrated that AT9283 promotes apoptosis and increases the population of cells in the G2/M phase (Bavetsias & Linardopoulos, 2015; Takeda et al., 2020). Our findings indicate that AT9283 exhibited dose- and time-dependent inhibition of BL cell lines and primary BL cells viability, as demonstrated in Fig. 1. Furthermore, AT9283 induced G2/M cell cycle arrest in Akata and P3HR1 cells by downregulating the expression of cyclin B1 protein (Fig. 2), a crucial regulator of the G2/M checkpoint (Eriksson et al., 2007).

Apoptosis, a programmed cell death mechanism, serves as a crucial process in curbing the proliferation of malignant tumors and suppressing the malignant phenotype of tumor cells (Su et al., 2015). The apoptosis pathway is currently a prime target for cancer treatment drugs. Effector caspases, which can selectively cleave proteins, play a pivotal role in most cellular apoptotic processes. Poly (ADP-ribose) polymerase-1 (PARP1), an enzyme that localizes in the nucleus, is closely associated with DNA repair under stress and is indispensable for cellular stability and survival (Satoh & Lindahl, 1992). In vitro, PARP can undergo cleavage and activation by various caspases, while in vivo, it serves as the primary substrate for caspase 3 cleavage (Lazebnik et al., 1994; Oliver et al., 1998). The significance of PARP lies in its crucial role in maintaining cellular stability and survival. Its cleavage accelerates cellular instability and is considered a significant indicator of apoptosis and Caspase 3 activation (Oliver et al., 1998). In this study, the impact of AT9283 on BL cell apoptosis was assessed through annexin V/PI double-staining flow cytometry and Western blotting. The activation of apoptosis was observed to be significant following treatment with AT9283, as evidenced by the increased protein levels of key apoptotic mediators Caspase 3 and cleaved PARP, which were found to be directly proportional to the concentration of AT9283 administered (Fig. 3).

The Warburg effect, characterized by abnormal metabolism, is a hallmark of cancer (Hanahan & Weinberg, 2011), with increased glucose consumption and lactate secretion being key features of many rapidly developing cancers. Consequently, targeting glycolysis in tumor cells represents a novel and promising therapeutic approach.The rate-limiting step in glycolysis is regulated by a collection of enzymes, such as HK2, PK. The enzyme is responsible for regulating the aerobic glycolysis pathway and directly influences gene expression and cell cycle regulation in tumor cells (Lu et al., 2018; Mazurek et al., 2005). Among these, HK2 plays a pivotal role as the key rate-limiting enzyme in the glycolytic pathway, facilitating glucose phosphorylation (Lis et al., 2016). Its overexpression is frequently observed in various types of cancers, indicating a poor prognosis (Anderson et al., 2017; Wilson, 2003; Wu et al., 2017). Additionally, HK2 has been identified as a crucial factor in the pathology of malignant B-cell lymphomas (Nakajima et al., 2019). While the mammalian genome encodes multiple isoforms of PK, cancer cells preferentially express a specific competitive subtype known as PKM2.

Numerous investigations have elucidated the crucial role of Aurora kinases, particularly Aurora A, in modulating glycolysis and oxidative energy metabolism (Cheng et al., 2019; Nguyen et al., 2021; Sun et al., 2020; Wang et al., 2020). A recent research showed that the Warburg effect is induced by Burkitt lymphoma (Looyens et al., 2021). Utilizing the GEPIA database, we acquired the expression levels of aerobic glycolysis-related kinases in both normal tissues and malignant B-cell lymphoma tissues. Our findings indicate that the expression of aurora kinases and the key proteins PKM2 and LDHA were significantly elevated in B-cell lymphoma tissues (Figs. 4A and 4B). We also analyzed the HPA database to evaluate the Aurora A and B mRNA levels in different cell lines, and revealed that the basal expression levels of Aurora A and B in lymphoma cell lines were high (Fig. S2). The results were confirmed by immunohistochemistry (Fig. 4C). The outcomes imply that malignant B-cell lymphoma exhibits elevated levels of aerobic glycolysis, and that aurora kinases play a role in regulating glycolysis.

c-Myc is another important molecule involved in the occurrence and development of Burkitt lymphoma and some types of DLBCL. The c-Myc protein plays a role invovling in cell proliferation, growth, apoptosis, metabolism, and other cellular processes (Fernandez, Sanchez-Arevalo & de Alboran, 2012). Further, c-Myc enhances the transcription of the genes encoding HK2 and PK, leading to accelerated aerobic glycolysis, or the Warburg effect. The Myc protein is the major modulator of the Warburg effect in BL cells (Mushtaq et al., 2015). Many studies indicated overexpression of AURKA is related to increased expression of c-Myc (Dauch et al., 2016; Lu et al., 2015; Yang et al., 2010), and inhibition of Aurora kinase A was found to downregulate c-Myc protein levels (Nguyen et al., 2021). However, c- Myc is not the only modulator of the genes involved in glucose metabolism. Many genes involved in the process are also direct targets of HIF-1. Both c-Myc and HIF-1 participate in tumor metabolism by regulating transportion of glucose, the TCA cycle, glycolysis and glutaminolysis in cancer cells (Chen & Russo, 2012). The cooperative action of c-MYC and HIF-1 facilitates the transcription of genes associated with aerobic glycolysis, including HK2, PKM2, LDHA, and ABC transporters, thereby augmenting the expression of the corresponding proteins and promoting the establishment of the Warburg effect, ultimately hastening tumor progression (Dang et al., 2008; Liu, Jin & Fan, 2021; Wen et al., 2022). The findings of a study have confirmed that the proteins HIF-1 and Myc work in collaboration to regulate a minimum of two genes that encode glycolytic enzymes (HK2 and PDK1) in cells of BL (Kim et al., 2007). Here, we found that inhibiting Aurora kinases with AT9283 significantly reversed the Warburg effect in Akata and P3HR1 BL cells, resulting in suppression of proliferation, G2/M arrest and induction of apoptosis. One of the the underlying mechanisms seems to involve the downregulating of the expression of the metabolism-related c- Myc and HIF genes. Considering the pivotal significance of these transcription factors, it is imperative for us to investigate the downstream effects they mediate in our forthcoming research.

P3HR1 cells were utilized as a model to investigate alterations in mRNA levels of related genes following treatment with AT9283. The findings from the RT-qPCR analysis revealed a significant reduction in the mRNA expression levels of aurora B, HK2, PKM2, and c-myc in cells subjected to AT9283 treatment, as compared to the control group (Figs. S3B–3E). These results are consistent with the observations obtained from Western blot analysis, indicating that the modulation of these proteins expression is, to some extent, regulated at the transcriptional level upon AT9283 treatment. However, the observed modifications in Aurora A and HIFα proteins do not align with the alterations in mRNA levels (Figs. S3A and S3F), suggesting that the regulation of Aurora A and HIF proteins by AT9283 operates via translational mechanisms rather than transcriptional processes. Our forthcoming inquiries will aim to elucidate the underlying mechanism accountable for the discrepancies in the expression of Aurora A and HIFα proteins and their corresponding mRNA molecules, thus establishing a basis for future research and progress in the field of AT9283 anti-BL.

Conclusions

In summary, our study presents novel experimental evidence demonstrating the efficacy of the Aurora kinase inhibitor AT9283 against Burkitt lymphoma through mechanisms involving the Warburg effect. These findings provide a basis for further research on AT9283 as a potential therapeutic agent for Burkitt’s lymphoma. Further investigations are required to ascertain the in vivo inhibitory effects of AT9283 on BL.

Supplemental Information

Supplemental Information 1 Dose–response curves of AT9283 in Akata and in P3HR1 cells, respectively

The dose–response curves of AT9283 in Akata and in P3HR1 cells were calculated using GraphPad sofware with the nonlinear regression curve fit.

Click here for additional data file.

Supplemental Information 2 Aurora A and B mRNA in lymphoma cell lines were high

Using the Human Protein Atlas (HPA, https://www.proteinatlas.org/) database, we evaluated the Aurora A and B mRNA levels in different cell lines.

Click here for additional data file.

Supplemental Information 3 Effect of AT9283 on the mRNA level of of related genes in P3HR1 cells

The cells were cultured with different concentrations of AT9283 for 48 h. Real-time PCR was performed to determine the mRNA level of related genes. The relative level was normalized to the value of β-actin. The results are expressed as fold changes compared to the control. The data represent the mean ± SD of three independent experiments performed in triplicate. *p < 0.05, **p < 0.01, ***p < 0.001 and ****p < 0.0001.

Click here for additional data file.

Supplemental Information 4 Raw data for Figure 1

Click here for additional data file.

Supplemental Information 5 Raw data for Figure 2

Click here for additional data file.

Supplemental Information 6 Raw data for Figure 3

Click here for additional data file.

Supplemental Information 7 Raw data for Figure 4

Click here for additional data file.

Supplemental Information 8 Raw data for Figure 5

Click here for additional data file.

Supplemental Information 9 Raw data for Figure 6

Click here for additional data file.

Supplemental Information 10 Raw data for Figure S3

Click here for additional data file.

Supplemental Information 11 Raw data

Click here for additional data file.

Supplemental Information 12 RT-PCR primers, RNA isolation and real-time PCR

Click here for additional data file.

The authors would like to thank the patient for participation and coordination in this study.

Additional Information and Declarations

Competing Interests

Author Contributions

Human Ethics

Data Availability

The authors declare there are no competing interests.

Kaiming Jiang conceived and designed the experiments, performed the experiments, analyzed the data, prepared figures and/or tables, and approved the final draft.

Lihong Bai performed the experiments, prepared figures and/or tables, and approved the final draft.

Canfei Wang performed the experiments, prepared figures and/or tables, and approved the final draft.

Xiang Xiao performed the experiments, prepared figures and/or tables, authored or reviewed drafts of the article, and approved the final draft.

Zhao Cheng analyzed the data, authored or reviewed drafts of the article, and approved the final draft.

Hongling Peng conceived and designed the experiments, analyzed the data, authored or reviewed drafts of the article, and approved the final draft.

Sufang Liu conceived and designed the experiments, performed the experiments, analyzed the data, prepared figures and/or tables, authored or reviewed drafts of the article, and approved the final draft.

The following information was supplied relating to ethical approvals (i.e., approving body and any reference numbers):

The Ethical Committee of the Second Xiangya Hospital of Central South University approved this study (Approval No. 2021772).

The following information was supplied regarding data availability:

The raw measurements are available in the Supplementary Files.

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
