# Peer review of "The Aurora kinase inhibitor AT9283 inhibits Burkitt lymphoma growth by regulating Warburg effect"

_PeerJ, doi:10.7717/peerj.16581_

## Round 0.1 · original submission · Major Revisions

· Academic Editor

Major Revisions

Please revise the manuscript according to the reviewers' remarks. Please pay particular attention to statistics.

Reviewer 1 ·

Basic reporting

General Comments
This manuscript provides an insightful investigation into the effects of AT9283, a kinase inhibitor, on Burkitt lymphoma (BL) cells. The study is well-conducted and robustly supported by data obtained through various methods including MTT assays, flow cytometry, western blotting, and immunohistochemical staining. The results could have substantial clinical implications for the treatment of Burkitt lymphoma. However, there are some areas where the manuscript could be improved for clarity, validity, and impact.

Specific Comments

Materials & Methods:

Please provide more information about the patient from whom the BMMNCs were isolated, including demographic details, to better evaluate the generalizability of the findings.
Statistical Analysis:

Results:

While the manuscript mentions a "dose-dependent" effect, there is no statistical analysis showing the dose-response curve fit. Consider adding this.


Minor Comments:

The abbreviation for Hexokinase is sometimes written as "HK2" and other times as "HK." Please be consistent.

Experimental design

No comments.

Validity of the findings

No comments.

Additional comments

No comments.

·

Basic reporting

This article primarily explores how the kinase inhibitor AT9283 affects Burkitt lymphoma by modulating the Warburg effect. Initially, the authors validate the phenotype showing that, post-AT9283 treatment, various BL cell lines are influenced through cell cycle arrest and apoptosis. They then identify several Warburg effect markers to demonstrate the influence of AT9283 on the Warburg effect. I commend the clarity of the experimental results which aptly support their hypothesis. Additionally, the references cited are both relevant and of high caliber.

However, there are areas where the article could benefit from refinements. For instance, in Fig. 1a, adjustments to the exposure for the Western Blot images would enhance clarity, and it would be beneficial to maintain a consistent format across all figures in terms of dimensions. Additionally, there's an inconsistency in the color scheme for the box figures; some are in grayscale while others are colored. It would be best to standardize this aspect. On line 203 of the manuscript, I believe the authors intended to reference "caspase 3", yet "caspase 9" is mentioned. For validating the Warburg effect, adding a scale bar to their IHC staining could enhance clarity. Lastly, inconsistencies like varying formats for "c-Myc" within a single paragraph need addressing. Attention to such details would elevate the overall quality of the manuscript.

Experimental design

Regarding the experimental design, I have several concerns I'd like to highlight:

The authors assess the expression levels of Aurora A and B in BL cell lines post AT9283 treatment. Yet, there's no mention of the basal expression levels of Aurora A and B in these cell lines or their significance to these cells. It would be beneficial to have supplementary data, possibly from sources like DepMap or CRISPR screens. Beyond just the protein level changes, I'd also suggest performing qPCR or bulk RNA-seq to validate changes at the mRNA level.
The authors posit that AT9283 triggers cell cycle arrest and apoptosis, substantiating this through flow cytometry. While the data seems to support their claims, the absence of both negative and positive controls is noticeable. Incorporating these controls would provide a more robust validation of their hypothesis.
As for the elucidation of the underlying mechanism, a deeper dive is required beyond merely indicating expression changes in c-Myc and HIF-1α. Given the critical role of these two transcription factors, it's essential to explore the downstream effects they mediate. Short-duration RNA-seq could elucidate downstream targets, and ATAC-seq might reveal genome-wide changes. From the presented data, the connection between these two proteins and the authors' hypothesis isn't clear.
These recommendations, if addressed, could significantly enhance the robustness and clarity of the study.

Validity of the findings

All data should undergo both biological and technical repetition at least twice to ensure accuracy. However, the authors have not addressed this aspect. Beyond the in vitro experiments, it's preferable to have in vivo data that corroborates the same hypothesis.

---

## Round 0.2 · accepted · Accept

· Academic Editor

Accept

Both reviewers now propose to accept your manuscript.

Reviewer 1 ·

Basic reporting

The authors have addressed all my concerns.

Experimental design

No comments.

Validity of the findings

No comments.

Additional comments

No comments.

·

Basic reporting

In light of the alterations introduced during the first revision, it is evident that the authors have adeptly elevated the scientific tenor of their discourse. The meticulous refinement of language employed imparts a heightened level of precision and clarity to their statements. This concerted effort has notably contributed to the overall comprehensiveness of the manuscript, fostering a more nuanced and insightful understanding of the subject matter. The authors' commitment to scientific rigor is palpable, as evidenced by the judicious selection of terminology and the enhanced coherence of the narrative. Such refinements not only bolster the scholarly merit of the work but also fortify its potential impact within the broader scientific community.

Experimental design

Following the thoughtful considerations articulated in both my comments and those of the fellow reviewer, it is discernible that the experimental design has undergone notable refinement. The incorporation of feedback has played a pivotal role in enhancing the overall cogency and rationale of the experimental framework. The adjustments made reflect a commendable responsiveness to the constructive critiques, resulting in a more methodologically robust and logically sound design. This iterative process of scrutiny and revision underscores the authors' commitment to methodological integrity and scientific rigor. The conscientious adaptation of the experimental design not only addresses specific concerns raised but also contributes substantively to the overall scientific merit of the study.

Validity of the findings

In response to both internal and external evaluative input, it is apparent that the authors have undertaken commendable efforts to augment the robustness of their findings. Through a meticulous revision process, they have systematically incorporated additional verification methods, thereby fortifying the foundation upon which their conclusions rest. This concerted endeavor to bolster the solidity of their research outcomes attests to the authors' dedication to scientific thoroughness and scholarly excellence. The diversified avenues of validation introduced subsequent to the revision not only serve to mitigate potential limitations but also engender a more comprehensive and defensible interpretation of the reported results. Such proactive engagement with constructive feedback underscores the authors' commitment to the methodological and empirical integrity of their scholarly endeavor.

Additional comments

While an augmented inclusion of in vivo data and sequencing information would undoubtedly fortify the evidentiary basis supporting the authors' proposition, it is noteworthy that the manuscript explicitly acknowledges this limitation and outlines a prospective course of action. The authors' transparency regarding their intentions to undertake further experimentation in subsequent phases of their research lends a sense of forward-looking responsibility. Given this articulated plan for future data acquisition, the current state of the manuscript can be reasonably considered within the context of an evolving research trajectory. While the prospect of additional in vivo and sequencing data is eagerly anticipated, the acknowledgment of this as a forthcoming avenue of investigation mitigates concerns about the current data landscape. The authors' conscientious acknowledgment of this limitation, coupled with their expressed commitment to expanding the dataset, serves to contextualize the current findings as part of an ongoing and progressively developing scientific inquiry.